# A systematic review and meta-analysis on the efficacy of glibenclamide in animal models of intracerebral hemorrhage

Tiffany F. C. Kung[1]*, Cassandra M. Wilkinson[1], Lane J. Liddle[1], Frederick Colbourne[1,2]

1 Department of Psychology, University of Alberta, Edmonton, Alberta, Canada, 2 Neuroscience and Mental Health Institute, University of Alberta, Edmonton, Alberta, Canada

* tkung@ualberta.ca

**Data Availability Statement:** All relevant data are within the paper and its Supporting Information files.

## Abstract

Intracerebral hemorrhage (ICH) is a devastating stroke with many mechanisms of injury. Edema worsens outcome and can lead to mortality after ICH. Glibenclamide (GLC), a sulfonylurea 1- transient receptor potential melastatin 4 (Sur1-Trpm4) channel blocker, has been shown to attenuate edema in ischemic stroke models, raising the possibility of benefit in ICH. This meta-analysis synthesizes current pre-clinical (rodent) literature regarding the efficacy of post-ICH GLC administration (vs. vehicle controls) on behaviour (i.e., neurological deficit, motor, and memory outcomes), edema, hematoma volume, and injury volume. Six studies (5 in rats and 1 in mice) were included in our meta-analysis (PROSPERO registration = CRD42021283614). GLC significantly improved behaviour (standardized mean difference (SMD) = −0.63, [−1.16, −0.09], n = 70–74) and reduced edema (SMD = −0.91, [−1.64, −0.18], n = 70), but did not affect hematoma volume (SMD = 0.0788, [−0.5631, 0.7207], n = 18–20), or injury volume (SMD = 0.2892, [−0.4950, 1.0734], n = 24). However, these results should be interpreted cautiously. Findings were conflicted with 2 negative and 4 positive reports, and Egger regressions indicated missing negative edema data (p = 0.0001), and possible missing negative behavioural data (p = 0.0766). Experimental quality assessed via the SYRCLE and CAMARADES checklists was concerning, as most studies demonstrated high risks of bias. Studies were generally low-powered (e.g., average n = 14.4 for behaviour), and future studies should employ sample sizes of 41 to detect our observed effect size in behaviour and 33 to detect our observed effect in edema. Overall, missing negative studies, low study quality, high risk of bias, and incomplete attention to key recommendations (e.g., investigating female, aged, and co-morbid animals) suggest that further high-powered confirmatory studies are needed before conclusive statements about GLC's efficacy in ICH can be made, and before further clinical trials are performed.

## Introduction

Intracerebral hemorrhage (ICH) accounts for 10–15% of all strokes, and has a devastating mortality rate of ~40% [1]. In ICH, a ruptured blood vessel leads to the formation of a

**Funding:** This research was supported by a project grant from the Canadian Institutes of Health Research Grant (#451168) to FC, CW and others. TK, LL, and CW were supported by Canadian Graduate Scholarship – Doctoral awards. CW was supported by an Izaak Walton Killam Memorial scholarship. The funders had no role in study design, data collection and analysis, decision to publish, or preparation of the manuscript.

**Competing interests:** The authors have declared that no competing interests exist.

parenchymal hematoma. This causes primary damage through mechanical trauma as blood dissects through brain tissue; secondary injury mechanisms occur following the initial bleed and include oxidative stress, neurotoxicity, edema, and many others [2].

In human ICH patients, cerebral edema rises rapidly in the first 24 hours, and peaks between 2–3 weeks post-ICH [3–5]. The majority of swelling develops around the area of the bleed, and is known as perihematomal edema (PHE) [5–8]. As early as 1 hour after ICH, clot retraction begins to stabilize the formed blood clot [5–8]; this process leads to serum protein extrusion into the surrounding area, which then creates an osmotic gradient and pulls fluid from the blood into the brain parenchyma, augmenting PHE [6]. Plasma-derived serum extrusion is a major source of edema after ICH [8], but many other processes contribute to PHE development.

Similar to serum extrusion, cytotoxic, ionic, and vasogenic edema all begin early after ICH [6,7]. In cytotoxic edema, the failure of active ion transporters results in $Na^+$ moving down osmotic gradients from the extracellular to the intracellular brain space via channels such as the sulfonylurea 1- transient receptor potential melastatin 4 (Sur1-Trpm4) channel [5,6,9–11]. Water follows, leading to pathological swelling, or cytotoxic edema [5,6,9–11]. Parallel to cytotoxic edema is ionic edema, which occurs in areas with an intact blood-brain barrier (BBB) [9]. The movement of $Na^+$ into the intracellular space through cytotoxic edema depletes the ion concentration in the extracellular brain space; this depletion then drives $Na^+$ to enter the brain from the blood, through channels such as Sur1-Trpm4 [5]. The influx of $Na^+$ into the brain's extracellular space creates an osmotic and hydrostatic pressure gradient for water to follow, leading to ionic edema, or swelling in the extracellular brain tissue [9,12]. Essentially, ions move from the blood into the extracellular space, and then into the intracellular space; the repetition of this process leads to cytotoxic and ionic edema [6,9]. Following damage to the BBB, vasogenic edema replaces ionic edema [5,9]. In vasogenic edema, the disrupted BBB means that plasma proteins like albumin enter the extracellular space of the brain, which creates a strong osmotic gradient for water, leading to tissue swelling [6,9,11,12]. Other contributors include thrombin, which peaks between 24–72 hours post-ICH in humans, and neurotoxicity from red blood cell breakdown products, which peaks >72 hours post-ICH in humans [5–7,13–17].

Edema plays an amplifying role for many other secondary damage mechanisms, which contributes to its potential as a treatment target. The ionic dyshomeostasis (notably of $Na^+$, $K^+$, and $Cl^-$) concomitant with edema undoubtedly impacts cellular health and electrophysiological function within the peri-hematoma zone [18–22]. Additionally, cerebral edema, coupled with the mass of the hematoma, causes midline shift and raises intracranial pressure (ICP), which can potentially lead to herniation and significant impairments in cerebral blood flow [23,24]. Although the relationship between edema and raised ICP is complex owing to the interplay of several compliance mechanisms [8,23,25,26], one may assume that reductions in edema should contribute to better recovery and lower mortality. Many animal studies indirectly support the beneficial effects of reducing edema on lowering ICP [6,11,27,28], but several animal and clinical studies do not [11,26,27,29–34]. A recent meta-analysis investigated the impact of PHE on ICH outcomes and found a weak association between PHE and clinical outcome [33,35], with high heterogeneity noted among studies. The high heterogeneity may be due in part to differences in PHE measurement among studies, as various methods of assessing PHE outcome (e.g., absolute vs relative edema, edema volume vs edema growth) and timepoints (e.g., 24 hours post-ICH, 72 hours post-ICH) were used [33,35]. While pre-clinical studies are more methodologically homogenous and predominantly use biochemical (wet-weight dry-weight) methods to assess edema, differences among study quality, sample sizes, and similar factors may have a large effect on differences among relatively homogenous animal

populations. Thus, there remains a need to better characterize the time course and pathophysiology of edema in both animals and humans to improve translation efforts [36]. Regardless, understanding the mechanisms of edema and testing anti-edema targets in ICH remains a possible treatment target worthy of investigation.

Glibenclamide (GLC) is used to treat Type II diabetes by blocking the Sur1 and Kir6.2 subunits of the adenosine triphosphate (ATP)-sensitive potassium ion channel [37]. In pre-clinical ischemic stroke studies, low doses of GLC, which do not typically cause hypoglycemia, have repeatedly been shown to reduce edema and improve mortality [38–47]. In stroke, GLC attenuates edema by blocking the Sur1-Trpm4 nonselective monovalent cation channel in the brain [41,45]. By blocking Sur1-Trpm4, GLC is thought to stop the transcapillary movement of $Na^+$ and accompanying water, preventing cytotoxic and ionic edema formation [9]. Specifically, GLC is thought to be most effective in cases of severe stroke, where edema is life-threatening [41]. However, human data on GLC's efficacy remains conflicted. A Phase II clinical trial in ischemic stroke found that GLC did not improve functional outcomes [47], but a follow-up analysis suggested that in patients 70 years of age or younger, GLC administration may reduce mortality [48]. A Phase III clinical trial is underway investigating GLC after large hemispheric infarctions [49]. Such a therapy is desperately needed as currently, thrombectomy and thrombolysis remain the only clinically approved therapies for ischemic stroke patients, which is limited in its eligible patient population [50].

Following these promising preliminary results in ischemic stroke, researchers have begun investigating GLC in other subtypes of stroke, including ICH. Importantly, although pathophysiological characteristics overlap between ischemic and hemorrhagic stroke, putative treatments that work in one subtype must still be rigorously assessed in the other. Indeed, there are many therapies that work in pre-clinical ischemic stroke models, but give far less impressive results in ICH (e.g., hypothermia [51]). Glibenclamide, which shows promise in ischemic stroke, may fall into this category. Currently, interest in GLC for the treatment of ICH stems from the general success of GLC in blocking cytotoxic and ionic edema in pre-clinical ischemic stroke models. However, unlike ischemic stroke models and patients investigated in GLC studies, serum protein extrusion appears as true edema in ICH, and is likely unaffected by GLC [6,8]; thus, original research on GLC specifically in the context of ICH is needed.

Although a variety of methods (e.g., minimally invasive surgery [52], therapeutic hypothermia [53]) are being investigated for treating edema, clinical guidelines currently recommend the use of osmotherapy (hypertonic saline (HTS), mannitol) for PHE and increased ICP in ICH [2,54,55]. Evidence for their overall efficacy in ICH remains conflicted (Evidence Level C) [2,54,56–58]; further, data are inconclusive on whether HTS or mannitol is safer or more effective [58–61], and both treatments remain commonly used [2,58–63]. Mannitol and hypertonic saline are both hyperosmolar treatments and diuretics, though through slightly different mechanisms [62]. Comparatively, glibenclamide has similar diuretic effects, but is a sulfonylurea and works by blocking the Sur1-Trpm4 channel, and putatively reducing transcapillary $Na^+$ flow [9]. Compared to HTS and mannitol, GLC is mainly being investigated for its effect on overall edema, rather than for the temporary management of ICP. While reviews evaluating the use of HTS and/or mannitol exist [61–63], no studies have directly compared GLC to hyperosmolar treatments, nor have any systematic reviews on GLC in ICH been performed.

Presently, most GLC work in ICH has been conducted in animal models. In rodents, edema peaks around 3 days post-ICH [16,28,64–66], compared to ~2–3 weeks in humans [5]. Researchers use two main methods of modelling ICH in animals, the autologous whole blood (AWB) model and the collagenase (COL) model. In the AWB model, whole blood is collected from the peripheral vasculature and infused into the desired brain region, usually the striatum [67,68]. In the COL model, bacterial collagenase is infused into the brain to break down blood

vessels, thereby mimicking spontaneous ICH [69]. Even when well matched for hematoma size, the AWB model results in less BBB damage, a smaller amount of edema, and a notably smaller increase in ICP compared to the COL model [22,23,26]. These significant differences in edema and related factors (i.e., BBB damage, ICP increases) mean the efficacy of GLC may differ between models. It is therefore crucial to synthesize pre-clinical evidence across both models to provide the most accurate and holistic perspective of GLC's actions in ICH and improve translational ability.

The vast majority of all biomedical therapies fail to successfully translate to the clinic, with no approved neuroprotective therapies existing for ICH [2,70]. Many studies have identified pitfalls that contribute to translational failures, including experimenter bias (e.g. failure to blind or randomize groups), the use of low sample sizes which lead to an overestimation of effect size, and a lack of diversity among investigated animals (e.g., a lack of female, aged, and co-morbid animals) [70–74]. High quality pre-clinical evidence bolsters the potential of successfully moving therapies across the translational valley [70,71,75]. However, the quality and content of pre-clinical evidence on GLC in ICH remains unexplored. Discrepancies among studies (e.g., positive vs negative findings, study quality) and current clinical interest in GLC reinforce the importance of understanding the pre-clinical literature. Thus, this meta-analysis synthesizes all currently available data on the impact of GLC (vs. vehicle controls) on outcomes (behavioural outcomes, edema, hematoma volume, injury volume) following experimental ICH in animal models.

## Materials and methods

This meta-analysis was pre-registered with PROSPERO (CRD42021283614) [76], and executed according to what was submitted. The PROSPERO pre-registration was written following Preferred Reporting Items for Systematic Reviews and Meta-Analyses (PRISMA) guidelines, and this manuscript adheres to the PRISMA 2020 guidelines for reporting data in meta-analyses [77]. A completed PRISMA checklist is available in the supporting information (S1 Appendix). The meta-analysis was conducted following the guide written by Vesterinen and colleagues [78]. For example, when control groups were compared to multiple treatment groups, group sizes were divided by the number of treatment groups it serviced, as was done for one study [79]. Inclusion and exclusion criteria are provided in Table 1, and in our pre-registered PROSPERO document (CRD42021283614) [76]. As the purpose of our meta-analysis was to evaluate the effectiveness of glibenclamide in treating patients with severe ICH, and not the efficacy in ICH patients already treated with GLC for other reasons, studies with a pre-ICH dosing administration were excluded for maximum translational relevance.

### Search terms

Our search was first conducted on March 15, 2022, on PubMed, EMBASE, CINAHL, SCOPUS, and MEDLINE using the following terms: ((Glibenclamide OR glyburide OR GLC or GLB or GLIB OR Diabeta OR Flycron) AND (cerebral hemorrhage OR cerebral haemorrhage OR intracerebral hemorrhage OR intracerebral haemorrhage OR intracranial hemorrhage OR intracranial haemorrhage OR ICH OR cerebral hematoma OR hemorrhagic stroke OR haemorrhagic stroke OR intraparenchymal hemorrhage OR intraparenchymal haemorrhage)) AND (experimental OR pre-clinical OR rat OR rats OR mouse OR mice OR monkey OR monkeys OR animal OR animals OR rodent OR rodents OR primate OR primates OR non-human OR animal model). Two subsequent searches on December 27, 2022, and April 23, 2023, were performed with the same search terms to identify any new literature that had been published following our first search.

**Table 1. Inclusion and exclusion criteria.**

|  | Inclusion Criteria | Exclusion Criteria |
|---|---|---|
| **Population** | All studies with experimentally induced ICH in animal models | *In vitro* studies and human clinical populations |
| **Intervention** | Any administration of GLC following initiation of ICH, regardless of administration route, dose, and treatment paradigm that was compared to a no-therapy group | Treatments that begin before initiation of ICH |
| **Comparator** | No-treatment control group. If multiple no-treatment groups are included, comparator will be selected based on the following hierarchy: 1) ICH + Vehicle 2) ICH + No Vehicle Group | No relevant control or comparator group present |
| **Outcome** | <u>Primary</u><br>Measures of beahvoioural outcome. If multiple assessments are performed, the latest assessment for early ($>$7 days) and late ($\leq$ 7 days) will be used. If multiple behavioural tests are performed per study, the average effect size will be used.<br><u>Secondary</u><br>Biochemical analyses of edema performed via wet-weight dry-weight, histological or imaging estimates as early as 1 hour post-stroke and as late as 5 days post-stroke.<br>Histological or biological assessments of hematoma volume as early as 1 day and as late as 3 days post-ICH.<br>Histological or biological assessments of injury volume as early as 1 day post-ICH. | Any study that did not investigate any of the detailed endpoints. |
| **Study Design** | Any controlled animal studies with an experimental group that undergoes post-ICH administration of GLC, with a separate control group | *In vitro* and human studies. Any animal design that has a pre-stroke administration of GLC. Case studies, cross-over studies, and studies without a separate control group. |
| **Other** | Full text must be available. If only an abstract is available, authors will be contacted by email for follow up information. If no response is received within 4 weeks, the study will be excluded. | Any study where full text is not available, or is not in English. |

Inclusion/exclusion criteria are also detailed in our pre-registered PROSPERO document [76].

## Screening process

Results from each search were loaded into Covidence for screening and data extraction [80]. Each study was independently screened by two reviewers (TK, CW, or FC), and a third reviewer was consulted in the case of differences (FC). Data extraction was completed by two independent reviewers (TK, CW, or FC). In cases of disagreement for data extraction, the original study was re-reviewed. In the case of missing data, lead authors were contacted via emails as provided on the manuscript, requesting clarifications. If no response was given after 4 weeks, data were manually extracted from the graphs in the original publication. Only two of six (33%) authors provided clarifications upon request, both of whom are authors on this meta-analysis.

Data were extracted for pre-defined endpoints, including all behavioural measures as our primary outcome. Measures of cerebral edema (brain swelling), and tissue damage (hematoma volume and injury volume) were extracted as our secondary outcomes. All other data were summarized qualitatively. Behavioural data were dichotomized into early ($\leq$7 days post-stroke) or late ($>$7 days post-stroke) assessments, and scores from the latest assessment time for each test were used. This dichotomy was chosen as edema in animal models begin to resolve around this time [28]. As an example, Wilkinson et al. investigated staircase at days 8–10 and 25–27 post-stroke. Data from days 25–27 (reported summarily rather than per day) were extracted for data synthesis.

Experimental design and study information was extracted. Information about study quality (e.g. blinding, randomization, *a priori* planning and exclusion criteria), animal information (rodent species, age/weight, and sex), experimental ICH information (COL or AWB, severity, and location), dosing information (timing of dosing in relation to ICH, length of dose administration, continuous or intermittent administration, route of administration), statistical analysis information, and study details (year of publication, declared conflicts of interest, funding sources, and country) were recorded. Studies were deemed eligible for extraction based on our pre-determined PROSPERO document.

## Study quality assessment

The CAMARADES (Collaborative Approach to Meta Analysis and Review of Animal Experimental Studies) checklist was used to assess study quality [81]. The checklist was slightly adapted by replacing instances of "ischemia" with "ICH" (e.g., "blinded induction of ischemia" was changed to "blinded induction of ICH"). Briefly, studies were rated as 'yes' or 'no' on 10 measures relating to study quality. The SYRCLE (SYstematic Review Centre for Laboratory animal Experimentation) risk of bias tool was used to screen for bias [82]. As recommended by the authors of the SYRCLE tool, the data were not transformed into scores, and are presented qualitatively. Briefly, studies were rated as low risk, unclear risk, or high risk of bias on 10 measures. For both scales, if certain items (e.g., method of randomization) were not clear from the manuscript, authors were contacted for clarification. If no response was received, the study was marked as 'unclear' or 'not known'. All studies were evaluated for risk of bias and study quality by two independent reviewers (TK, CW) and disagreements were settled by re-review and discussion.

## Statistical analysis

Mean and standard deviation were extracted from all studies for all endpoints. Non-parametric data were assumed to be parametric to allow for combination with parametric data. Many studies inappropriately analyzed non-parametric data with parametric statistics, meaning parametric statistics were more readily available. Since raw data were not available for all studies, conversion to more fitting statistics was not possible. All data were extracted and analysed as standardized mean differences (SMD) [83] for all endpoints, due to missing sham or baseline data among studies. Hedge's G was used for all effect sizes due to the small number of studies in the analysis, as recommended [78].

Due to apparent differences between studies, a random-effects meta-analysis model was used with Dersimonian-Laird estimators. Data were analysed using R (ver. 4.2.1) using the metafor and meta packages [84–87]. Statistical heterogeneity was assessed using the $I^2$ statistic. Egger regressions and funnel plots were conducted to assess possibility of publication bias. Meta-regression was used to explore heterogeneity in endpoints assessed by 5 studies (k = 5) or more. Due to the low number of studies in the meta-analysis, only one meta-regression factor was explored per outcome. Sensitivity analyses were conducted when outlying points that may contribute disproportionately to the model were observed.

## Outcomes

Behavioural assessments were set as our primary endpoint, as they are a critical translational and clinical factor to understanding how treatments may impact quality of life in human patients [73]. All studies but one [88] assessed behavioural outcomes. All studies that assessed behaviour did so with more than one test [79,89–92]. Behavioural tests used, times assessed

(early or late), assessment time in relation to ICH and dosing, and raw scores or change from baseline scores were extracted from studies as available.

All studies performed edema analyses. For this secondary outcome, the volume of the brain assessed, assessment type (histological, biochemical, or by imaging), inclusion of the hematoma in analysis, and bleed size (if known) was extracted.

Only three studies conducted hematoma or injury volume assessments [88,89,92]. For these secondary outcomes, the assessment type (histological, biochemical, or by imaging), time of assessment, and brain volume assessed (e.g., entire hemisphere, lesion only) were extracted.

## Results

### Search results

We identified 516 potentially relevant articles in our searches (March 2022, December 2022, April 2023). Of these, 505 abstracts were omitted for being irrelevant, and 5 full studies were excluded after full-text screening, 4 for incorrect patient population/model (hemorrhagic transformation [42,93], traumatic brain injury [94], human population [95]) and 1 for an excluded dosing regimen (pre-ICH dosing regimen [96]) (Fig 1). This left 6 studies for data analysis and extraction.

### Study characteristics

For the purposes of this review, stroke severity was estimated using hematoma volume for COL-induced ICH, and volume of injected blood for AWB-induced ICH. Stroke severity and additional relevant study details are provided in Table 2.

### Primary outcome—behaviour

All studies but one performed behavioural assessments (Table 3), with three studies investigating both early ($\leq$7 days) and late (>7 days) behaviour [89–91], and two studies investigating early behaviour only [79,92].

Our random-effects meta-analysis showed that GLC significantly improved behavioural scores (SMD = −0.6281 with 95% CI [−1.1622, −0.0939], p = 0.0212; Fig 2A). Our analysis also indicated moderate heterogeneity ($I^2$ = 48.33%, p = 0.1015). A trim-and-fill analysis indicated no significant evidence of publication bias (Egger regression p = 0.0766; Fig 2B). A sub-group analysis found that dichotomized (early vs late) timepoints had no predictive value on GLC's effects on behaviour (p = 0.7006).

### Secondary outcome–edema

All studies performed edema measurements and used wet-weight dry-weight measurements of edema (Table 3). Four studies assessed edema at 72 hours post-stroke [79,89–91], one study at 24 hours [88], and one study at both 24 hours and 72 hours [92].

Our random-effects model indicated a significant effect of GLC on edema, favouring treatment (SMD = −0.9088 with 95% CI [−1.6384, −0.1793], p = 0.0146; Fig 3A). However, our analysis also indicated high heterogeneity ($I^2$ = 69.11%, p = 0.0063). A trim-and-fill analysis and Egger regression indicated statistically significant publication bias and missing negative studies (Egger regression p = 0.0001; Fig 3B). A meta-regression found no predictive value of stroke severity on GLC's effects on edema (p = 0.7252).

Due to the presence of an apparent outlier [79], a sensitivity analysis was performed. Following removal of the outlier, the effect on edema was still significant (SMD = −0.7065 with 95% CI [−1.3065, −0.1065], p = 0.0210; Fig 4A). Heterogeneity was reduced, though it

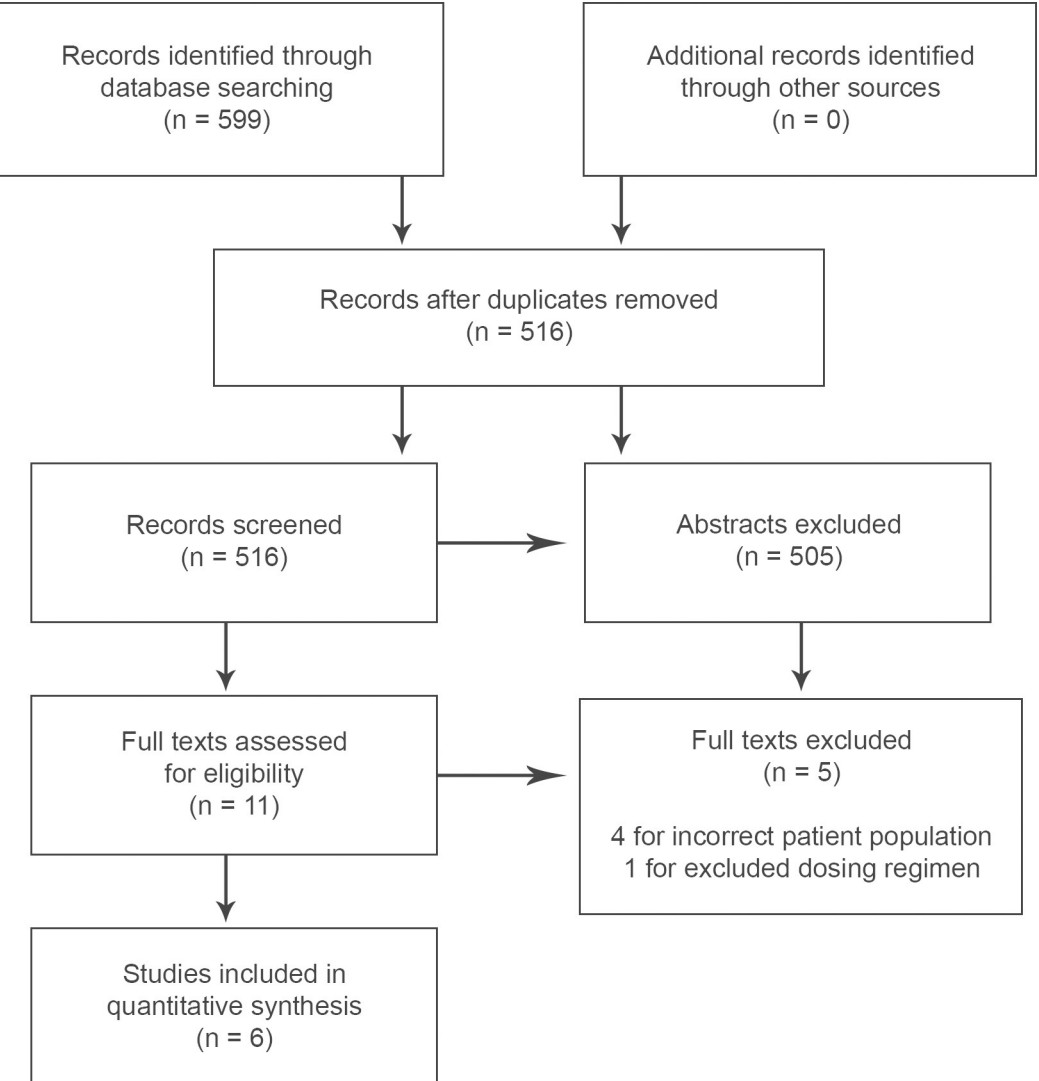

**Fig 1. PRISMA flow diagram.** A flow diagram for the meta-analysis is provided. Of the five full texts excluded, four were excluded for inappropriate patient populations/models [42,93–95], and one for using an excluded dosing regimen [96].

remained significant ($I^2$ = 59.05%, p = 0.0445). Evidence for missing negative studies persisted (Egger regression p = 0.0024; Fig 4B), and stroke severity remained a poor moderating variable (p = 0.1861).

### Secondary outcome–hematoma volume

Three studies performed hematoma volume measurements at 24 hours post-ICH, all of which used COL-induced ICHs [88,89,92]. Two studies assessed hematoma volume biochemically [89,92], and one study assessed it histologically (2 mm thick sections) [88].

The random-effects meta-analysis showed no significant effect of GLC on hematoma volume (SMD = 0.0788 with 95% CI [−0.5631, 0.7207], p = 0.8099; Fig 5A), and low statistical heterogeneity ($I^2$ = 0%, p = 0.8495). Due to the low number of studies, no further meta-bias or meta-regression analyses were performed.

**Table 2. Study characteristics.**

| Study | Jiang 2017 [91] | Xu 2019 [79] | Wilkinson 2019 [89] | Kung 2021 [92] | Jiang 2021 [90] | Shiokawa 2022 [88] |
|---|---|---|---|---|---|---|
| Country | China | China | Canada | Canada | China | Japan |
| Animal Species | Sprague Dawley rats | C57/Bl6 mice | Sprague Dawley rats | Sprague Dawley rats | Sprague Dawley rats | Sprague Dawley rats |
| Animal Sex | Male | Male | Male | Male | Male | Male |
| Animal Age / Weight | 250–300 g adults | 20–25 g adults | 275–400 g adults | 300–400 g adults | 450–550 g older adults | 237–388 g adults |
| ICH Model | AWB | AWB | COL | COL | AWB | COL |
| ICH Severity | 100 µL | 30 µL | 15 µL | 40/90 µL | 100 µL | 90 µL |
| GLC Initiation (post-ICH) | 0 h | 0 h / 2 h | 2 h | 2 h | 0 h | NK |
| GLC Dose | 10 µg/kg loading, 200 ng/h continuous | 10 µg single dose | 10 µg/kg loading, 200 ng/h continuous | 10 µg/kg loading, 200 ng/h continuous | 10 µg/kg loading, 200 ng/h continuous | 1 µL/h continuous |
| GLC Route | IP loading, SC continuous | IP | IP loading, SC continuous | IP loading, SC continuous | IP loading, SC continuous | SC |

Relevant study characteristics are summarized. For ICH severity, the amount of blood injected was used for studies employing the autologous whole blood (AWB) model. Results from hematoma volume analyses were used for studies employing the collagenase (COL) model. Of note, Jiang et al., 2022 [5] used older adult rats, equivalent to ~30–45 year old humans [97]. IP = intraperitoneal, SC = subcutaneous, NK = Not known.

## Secondary outcome–injury volume

Two studies investigated injury volume histologically, both coming from the same lab [89,92]. One assessed injury volume at 24 hours post-stroke [92], and one at 28 days [89]. Our random-effects meta-analysis showed no significant effect of GLC on injury volume (SMD = 0.2892 with 95% CI [−0.4950, 1.0734], p = 0.4699; Fig 5B), with no significant statistical heterogeneity ($I^2$ = 36%, p = 0.2112). Due to the low number of studies, meta-bias and meta-regression analyses were not conducted.

## Study quality

Risk of bias was assessed using the SYRCLE tool [82]. Overall, the risk of bias was unclear or high for most measures, particularly for random housing, random outcome assessment, and incomplete outcome data (Table 4). Study quality assessed using the CAMARADES checklist [81], was overall low for blinding, anesthetic choice, animal model, and sample size calculation (Table 5). In our meta-analysis, negative studies tended to have lower risks of bias (Table 4) and higher overall study quality (Table 5).

**Table 3. Methodological details.**

| Study | Behavioural Tests | Edema |
|---|---|---|
| **Jiang 2017 [91]** | Morris Water Maze, modified Neurological Severity Score | Bilateral 4 mm thick section |
| **Xu 2019 [79]** | Garcia test, Rotarod | Bilateral section, unknown thickness |
| **Wilkinson 2019 [89]** | Neurological Deificit Score, Montoya Staircase, Ladder test | Bilateral 6 mm thick section |
| **Kung 2021 [92]** | Neurological Deficit Score, Forelimb Placing test | Bilateral 6 mm thick section |
| **Jiang 2021 [90]** | Morris Water Maze, Forelimb Placing test, Corner Turn Test | Bilateral 4 mm thick section |
| **Shiokawa 2022 [88]** | N/A | Ipsilateral 2 mm anterior section |

Behavioural assessments varied across studies, as did method of measuring edema, as shown.

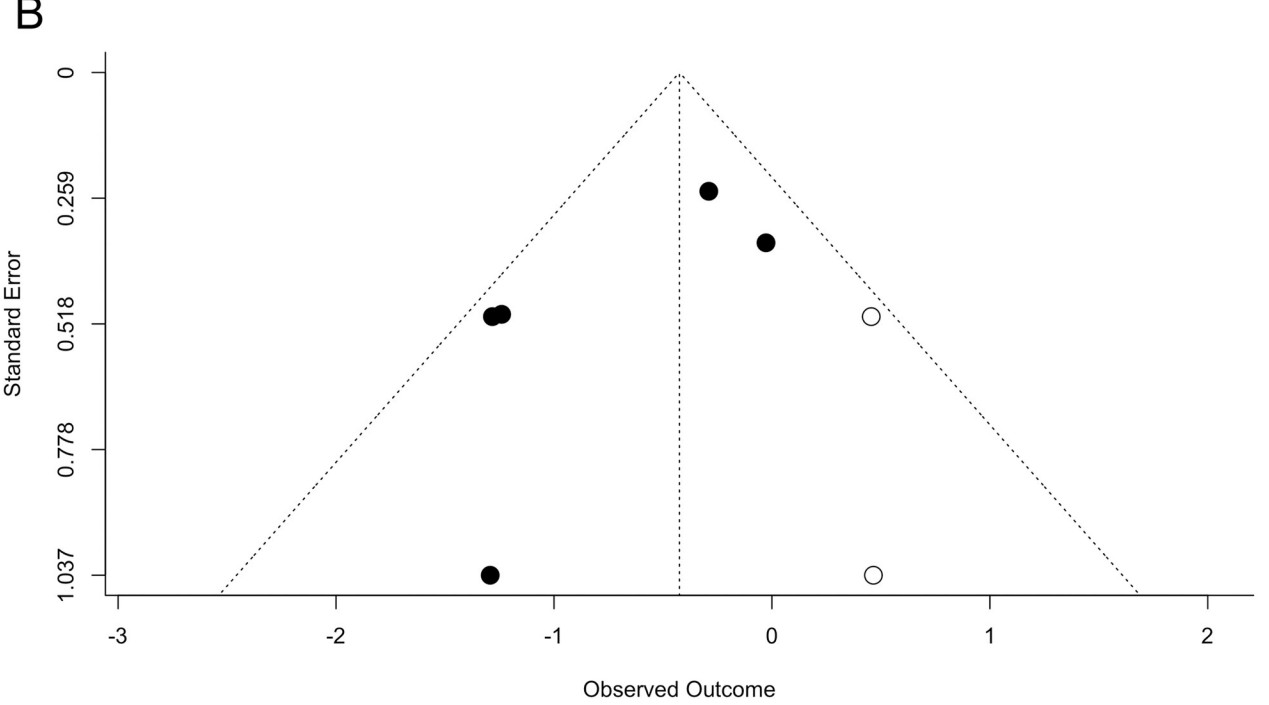

**Fig 2. Quantitative analysis of behaviour.** Meta-analysis was performed on all studies that assessed behavioural endpoints using a random-effects meta-analysis. (A) Glibenclamide significantly improved behavioural outcomes (p = 0.0212). (B) A trim-and-fill analysis primarily filled in the right side of the graph, indicating missing negative studies, but an Egger regression did not indicate publication bias (p = 0.0766), though this should be interpreted with caution considering our small sample size.

## Statistical analyses

No studies tested assumptions before using parametric tests. One study elaborated on adjustments made for unequal variances, but did not provide results for other assumptions [88]. Only three studies performed power calculations to determine sample sizes [79,89,92], though

## A

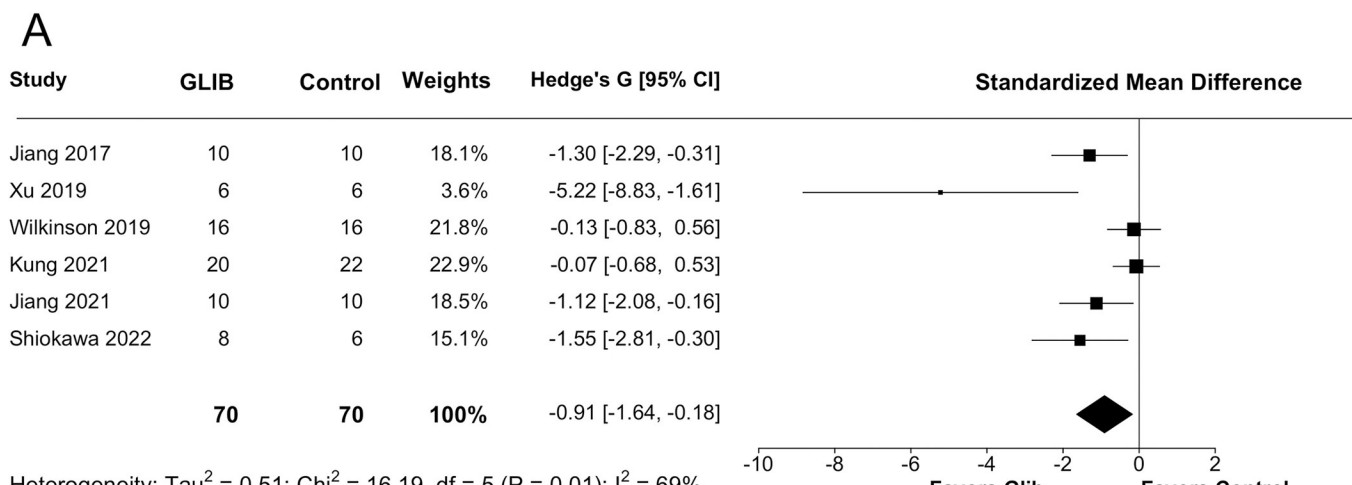

| Study | GLIB | Control | Weights | Hedge's G [95% CI] |
|---|---|---|---|---|
| Jiang 2017 | 10 | 10 | 18.1% | -1.30 [-2.29, -0.31] |
| Xu 2019 | 6 | 6 | 3.6% | -5.22 [-8.83, -1.61] |
| Wilkinson 2019 | 16 | 16 | 21.8% | -0.13 [-0.83, 0.56] |
| Kung 2021 | 20 | 22 | 22.9% | -0.07 [-0.68, 0.53] |
| Jiang 2021 | 10 | 10 | 18.5% | -1.12 [-2.08, -0.16] |
| Shiokawa 2022 | 8 | 6 | 15.1% | -1.55 [-2.81, -0.30] |
| | **70** | **70** | **100%** | -0.91 [-1.64, -0.18] |

Heterogeneity: $Tau^2 = 0.51$; $Chi^2 = 16.19$, df = 5 (P = 0.01); $I^2 = 69\%$
Test for overall effect: Z = -2.44 (P = 0.01)

## B

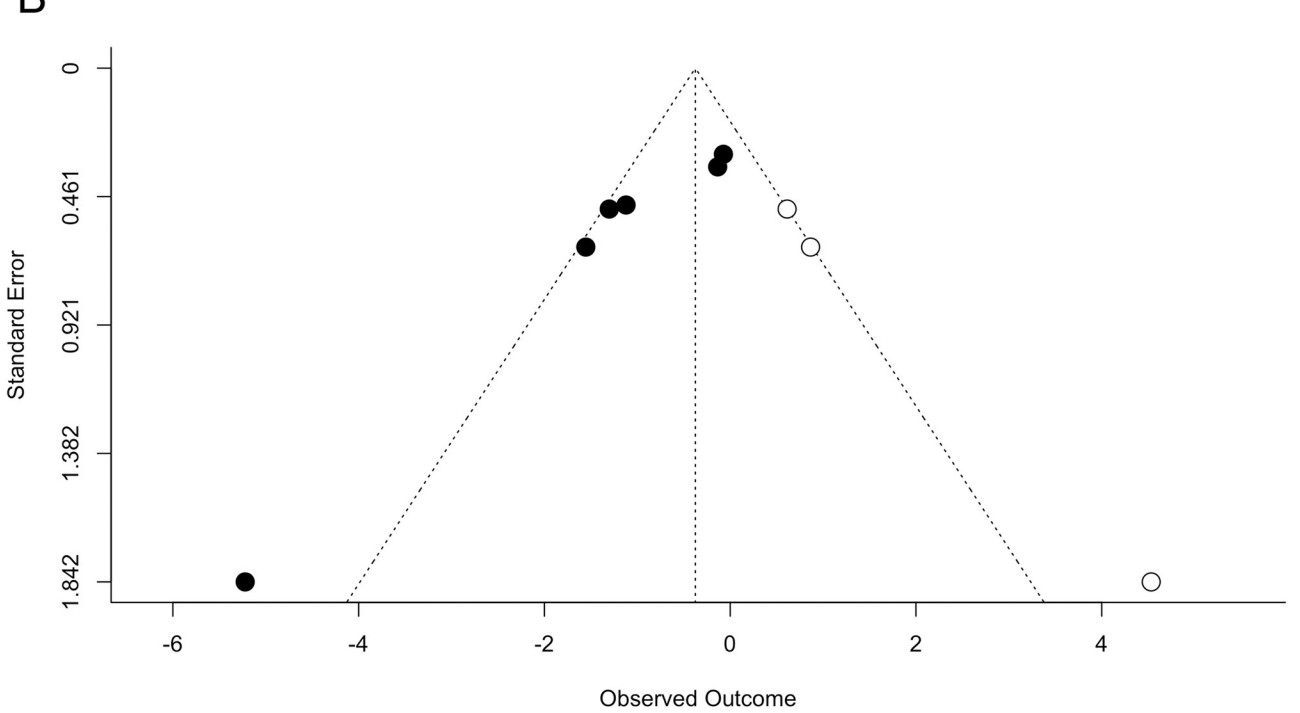

**Fig 3. Quantitative analysis of edema.** Meta-analysis was performed on all studies that assessed behavioural endpoints using a random-effects meta-analysis. (A) Glibenclamide significantly reduced edema (p = 0.0146). (B) A trim-and-fill analysis primarily filled in the right side of the graph, and an Egger regression indicated significant publication bias (p = 0.0001), suggesting missing negative data.

one did not provide expected effect sizes [79]. Three studies inappropriately used parametric statistics to analyze ordinal behaviour data [79,90,91].

### Other outcomes

Each study examined additional outcomes that were not among our pre-determined ones. These are briefly summarized below.

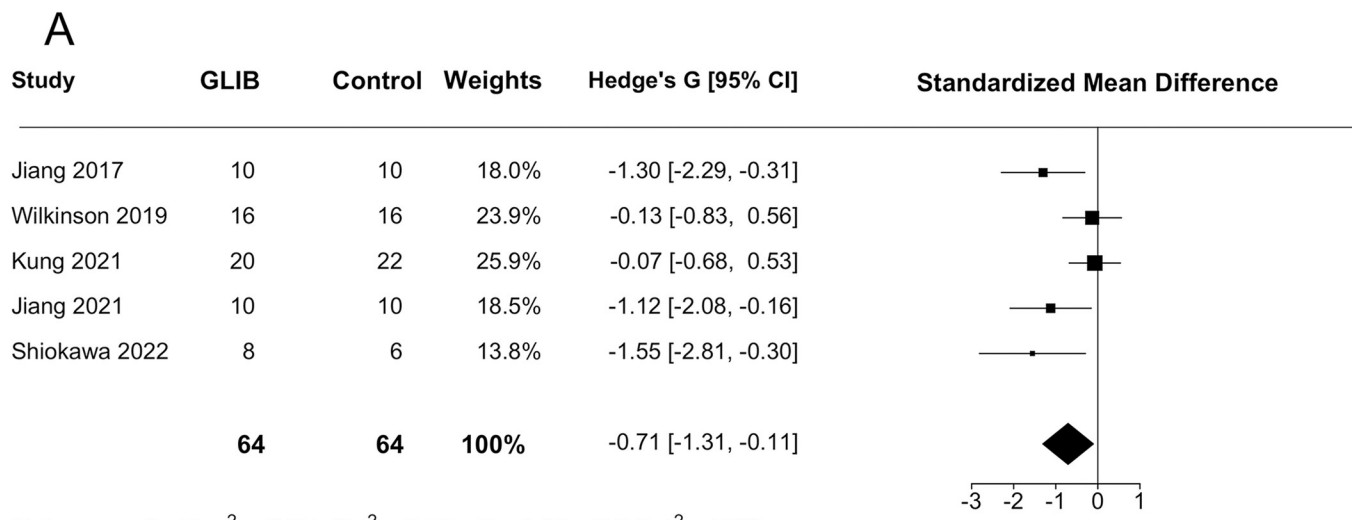

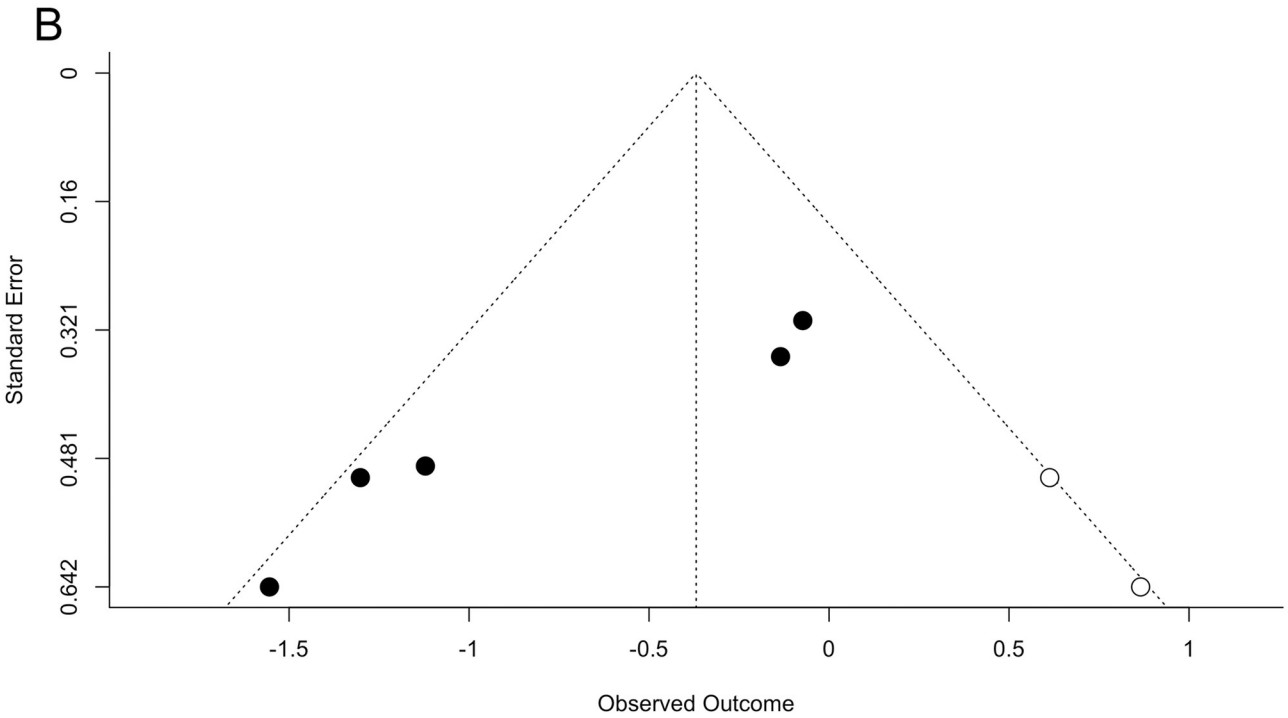

**Fig 4. Sensitivity analysis of edema.** A sensitivity analysis was performed on edema results after removing one study from the meta-analysis. (A) Glibenclamide significantly reduced edema (p = 0.0210), though effect size decreased from Hedge's G = −0.91 to −0.71. (B) A trim-and-fill graph of the sensitivity analysis still indicated publication bias (p = 0.0024).

**Safety measures.** Four studies assessed the effect of GLC on blood glucose [88,89,91,92], three of which did not find any effect of GLC [88,89,92], and one of which found decreased blood glucose on days 1–7 post-stroke [91]. It is unknown if these decreases were statistically significant.

Two studies assessed safety measures associated with GLC administration [76,89]. GLC did not affect temperature, though mild hyperthermia was observed in all groups post-ICH [76].

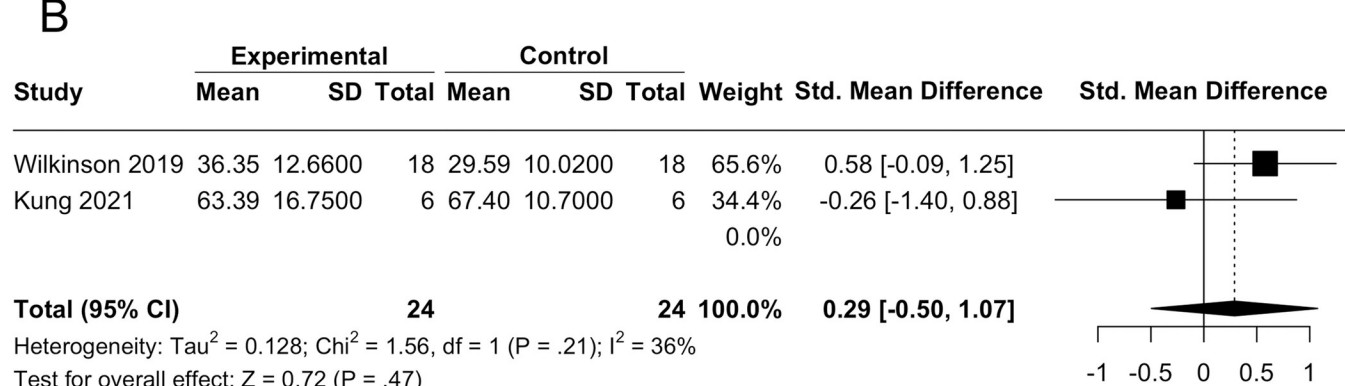

**Fig 5. Quantitative analyses of hematoma and injury volume.** Meta-analyses were performed on all studies that assessed hematoma and injury volume. (A) GLC did not affect hematoma volume (p = 0.8099). (B) No effect of GLC was found on injury volume (p = 0.4699).

Activity and food and water intake were similarly decreased post-ICH, but were not affected by GLC [76,89]. Pain did not differ by group [92].

**Channel expression.** Four studies investigated the expression of genes associated with the Sur1-Trpm4 channel (88,90–92). ICH upregulated *Abcc8* (codes for Sur1) gene expression and Sur1 protein expression without affecting Kir6.2 protein expression, though Sur1 upregulation was not affected by GLC administration [88,90,91]. The upregulation of Sur1 seemed to be localized to neurons and endothelial cells [91]. Data were conflicted, as one study did not find any differences in *Abcc8* or *Trpm4* mRNA expression between sham and ICH [76].

**Blood-Brain Barrier (BBB) integrity.** Four studies investigated BBB integrity [79,89,91,92]. Using bilateral spectrophotometric analyses and immunofluorescent histology of Evan's Blue extravasation, two studies found that GLC reduced BBB permeability, and increased tight junction (ZO-1 and Occludin) expression [79,91]. Using gadolinium penetrate (Magnevist®) injections analyzed with inductively coupled plasma mass spectrometry, two studies found no impact of GLC on BBB integrity [89,92].

**Inflammation.** Four studies investigated neuroinflammation [79,88,90,91]. GLC significantly reduced the inflammatory response by decreasing MMP-2 gene expression [91], IκκB expression in microglia [90], NF-κB p-65, IL-1ß, IL-18, TNFα, IL-1 protein expression [79,90], and galectin-3 and CD11b protein expression [88]. GLC-treated animals also displayed fewer ameboid-type microglia and more ramified microglia [88]. MMP-9 and MMP-12 were not affected by GLC administration [91], and IL-6 results were conflicted [79,90].

**Table 4. Risk of bias among included studies.**

| | Jiang 2017 [91] | Xu 2019 [79] | Wilkinson 2019 [89] | Kung 2021 [92] | Jiang 2021 [90] | Shiokawa 2022 [88] |
|---|---|---|---|---|---|---|
| **Sequence Generation** | *Low Risk* | **High Risk** | *Low Risk* | *Low Risk* | *Low Risk* | *Low Risk* |
| **Baseline Characteristics** | Unclear Risk | Unclear Risk | *Low Risk* | *Low Risk* | Unclear Risk | *Low Risk* |
| **Allocation Concealment** | Unclear Risk | **High Risk** | *Low Risk* | *Low Risk* | Unclear Risk | Unclear Risk |
| **Random Housing** | Unclear Risk | Unclear Risk | Unclear Risk | Unclear Risk | Unclear Risk | Unclear Risk |
| **Blinding Intervention** | **High Risk** | **High Risk** | *Low Risk* | *Low Risk* | **High Risk** | **High Risk** |
| **Random Outcome Assessment** | Unclear Risk | Unclear Risk | Unclear Risk | Unclear Risk | Unclear Risk | Unclear Risk |
| **Blinding Outcome** | **High Risk** | Unclear Risk | *Low Risk* | *Low Risk* | **High Risk** | *Low Risk* |
| **Incomplete Outcome Data** | Unclear Risk | Unclear Risk | *Low Risk* | *Low Risk* | Unclear Risk | **High Risk** |
| **Selective Outcome Reporting** | **High Risk** | **High Risk** | *Low Risk* | *Low Risk* | *Low Risk* | *Low Risk* |
| **Other Sources of Bias** | *Low Risk* | **High Risk** | *Low Risk* | *Low Risk* | *Low Risk* | *Low Risk* |

The SYRCLE risk of bias demonstrated generally unclear to high risk of bias. Only two lead authors (Wilkinson et al., 2019 [89]; Kung et al., 2021 [92]) indicated that animals were randomly housed (low risk for random housing, not depicted); others did not respond to email contact.

**Ion concentrations.** Two studies assessed striatal ion concentrations using inductively coupled plasma mass spectrometry [89,92]. Increased $Na^+$ and decreased $K^+$ concentrations after ICH were both associated with increased edema and greater BBB permeability [89]. GLC did not affect $Na^+$, $K^+$ (concomitant with edema) or Fe (rough estimator of hematoma volume) concentrations [89,92].

**Table 5. Study quality among included studies.**

| | Jiang 2017 [91] | Xu 2019 [79] | Wilkinson 2019 [89] | Kung 2021 [92] | Jiang 2021 [90] | Shiokawa 2022 [88] |
|---|---|---|---|---|---|---|
| **Peer Reviewed Publication** | *Yes* | *Yes* | *Yes* | *Yes* | *Yes* | *Yes* |
| **Control of Temperature** | *Yes* | **No** | *Yes* | *Yes* | *Yes* | **No** |
| **Random Allocation to Treatment or Control** | *Yes* | **No** | *Yes* | *Yes* | *Yes* | *Yes* |
| **Blinded Induction of ICH** | **No** | **No** | *Yes* | *Yes* | **No** | **No** |
| **Blinded assessment of outcome** | mNSS only | Behaviour only | *Yes* | *Yes* | Corner & FLP only | *Yes* |
| **Use of anesthetic without significant intrinsic neuroprotective activity** | **No** | NK | **No** | **No** | **No** | **No** |
| **Animal model (aged, diabetic, hypertensive)** | **No** | **No** | **No** | **No** | Middle Aged | **No** |
| **Sample Size calculation** | **No** | *Yes* | *Yes* | *Yes* | **No** | **No** |
| **Compliance with animal welfare regulations** | *Yes* | *Yes* | *Yes* | *Yes* | *Yes* | **No** |
| **Statement of potential conflicts of interest** | *Yes* | **No** | *Yes* | *Yes* | *Yes* | *Yes* |

The CAMARADES checklist was used to assess study quality. mNSS = modified Neurological Severity Score, FLP = Forelimb placing test, Corner = Corner turn test, NK = Not Known.

**Cell volume and density.** One study assessed cell volume and density via unbiased stereology at 24 hours post-ICH [92]. GLC significantly reduced cell density and increased cell volume in ipsilateral S1. Cell volume and density in contralateral S1, CA1, and CA3 did not differ.

**BDNF expression.** One study found that GLC significantly increased ipsilateral neuronal BDNF expression, as assessed via western blot (72 h post-ICH) and immunofluorescence (24 h post-ICH) [90].

## Discussion

Our meta-analysis found that GLC, when administered between 0 to 2 hours after ICH ictus, reduced edema and improved behavioural outcomes, without affecting bleeding or lesion volume. Importantly, the small number of studies in the field and concerns with study quality suggest that these results should be interpreted with caution. For example, although GLC improved outcomes, the estimated effect size had large 95% CIs of [−1.2 to −0.1] SMD for behaviour and [−1.6 to −0.2] SMD for edema, as well as high statistical heterogeneity ($I^2$ = 48–69%). Egger regressions also indicated significant publication bias and missing negative studies for edema (p = 0.0001) and trended towards significance in behaviour (p = 0.08). Lastly, our risk of bias and study quality assessments suggest that the studies included in our meta-analysis demonstrate issues surrounding bias and quality. Notably, only two studies blinded intervention, both of which were negative studies [89,92], and only three studies blinded all outcomes [88,89,92]; two studies failed to blind subjective behavioural analyses [90,91]. Further, two studies failed to report all mentioned outcomes [79,91]. The wide range in treatment effect sizes, the fact that several studies failed to find benefit of GLC, and the limited number of studies, all result in our low confidence in the current literature with respect to predicting whether and by how much GLC may impact outcomes in ICH patients. Prior to further clinical study, more data from high-powered confirmatory studies are needed to better understand what the true effect sizes may be [98,99].

More than 99% of all therapies fail to translate from pre-clinical studies to humans [70]. In order to prevent the continuance of this trend, the Animal Research: Reporting of In Vivo Experiments (ARRIVE) guidelines have called for researchers to improve their scientific rigor [100,101]. A recent study by Liddle et al. noted that despite these efforts, overall study quality remains poor in the ICH literature [74]. As mentioned above, we similarly found that the included studies demonstrate relatively high risks of bias (SYRCLE) and overall low study quality (CAMARADES). Additionally, ARRIVE guidelines were poorly followed, despite the translational nature and novelty of the field. Sample sizes, for example, varied greatly across studies, with both negative studies [89,92] employing larger sample sizes than positive studies [79,88,90,91]. Honest, *a priori* power analyses are important to prevent the overestimation of effect sizes, which decreases replicability and clinical translation [70,74]. In our meta-analysis, only three studies performed power analyses (2 negative reports, 1 positive report) [79,89,92], which resulted in varied sample sizes from n = 3 to n = 20 per group, indicating a wide range of expected effect sizes. Studies that did not perform power analyses ranged from n = 6–10 per group [88,90,91]. Future studies should employ group sizes of n = 33 to investigate edema, and n = 41 to investigate behaviour; our findings suggest these group sizes will provide 80% power to detect our observed effect sizes (Hedge's G = 0.9088 for edema, G = 0.6821 for behaviour). For reference, the largest study in our meta-analysis had pooled sample sizes of n = 20–22 in edema and n = 32–24 in behaviour [92], with most studies averaging far below that. Notably, group sizes of n = 20 are unusual in the ICH field, with most studies averaging around n = 6 [74].

To specifically aid in clinical translation in the stroke field, the Stroke Therapy Academic Industry Roundtable (STAIR) guidelines were updated in 2009 to guide researchers on steps that should be taken prior to clinical work [102]. The STAIR guidelines include recommendations to test therapies in multiple demographics (i.e., male and female, comorbid, aged animals), multiple species (e.g., rodents, non-human primates), and fully characterize the drug's pharmacodynamic profile (e.g., therapeutic window, etc.). Currently, GLC has been tested in the two most commonly used models of ICH (COL and AWB) and has been investigated in both rats and mice. However, many STAIR requirements remain to be fulfilled, suggesting further pre-clinical work is needed before continuing clinical investigations. All extant studies have been exploratory, not confirmatory, which partly explains the limited demographic of animals that have been studied; all studies used male rodents, five of which used Sprague Dawley rats, and one of which used C57/Bl6J mice. Only one study investigated middle aged rats (450–550 g, equivalent to ~30–45 years of age in humans) who still fall below the estimated average age of ICH patients [97,103], while the rest all investigated young adult rats. GLC has not been studied in female animals, aged animals, animals with co-morbid health conditions, or in other species (e.g., non-human primates). In addition to diversifying the study population assessed, further pre-clinical work on GLC's therapeutic window in ICH is needed. In our meta-analysis, five studies administered GLC within 2 hours post-stroke, while one study did not provide dosing timepoints. The dosing timeline used in the included studies may limit clinical relevance, as patients often do not arrive to the hospital and get diagnosed with ICH within this timeframe [104]. Further, in COL-induced ICH models, bleeding is ongoing for hours, meaning that GLC is likely acting while the bleed is still actively developing [105]. Such a protracted course of bleeding may occur in patients, but it is less common (~13–38% of patients) [106,107]. Though pre-clinical studies have indicated an extended 10 hour therapeutic window following ischemic stroke [38], the same has not been assessed for ICH. Lastly, other recommendations from STAIR guidelines remain to be performed (e.g., there are currently very limited dosing data and no dose-response curve data). These steps are critical to ensure the drug will benefit a clinically representative population without adverse side effects.

Further complicating the potential clinical translation of GLC are differences between rodents and humans that warrant careful consideration. PHE in humans occurs mostly in the white matter of the brain [5,108]. However, rodents display different white matter development and white-to-gray matter ratios than humans, and these discrepancies may limit the ability of rodents to faithfully model edema development in humans [109,110]. Additionally, more research is needed to understand the seemingly robust effect of GLC in AWB models of ICH. As a large extent of edema in the AWB model is attributable to serum extrusion, which GLC should not impact, the significant positive findings in the model suggest that a deeper mechanistic understanding of GLC is needed. As studies were not sufficiently powered, there remains a possibility that these results may have been false positives. Lastly, data regarding the use of GLC in ICH may be relevant to ischemic stroke in the context of hemorrhagic transformation.

Although existing pre-clinical data suggest the need for further pre-clinical research, a recent clinical study from China nonetheless investigated the efficacy of GLC in patients who experienced mild striatal ICH in the Glibenclamide Advantage in Treating Edema after ICH (GATE-ICH) study [111]. Glibenclamide (1.25 mg) was administered orally, 3 times daily, beginning on average between 22–24 hours after stroke. The timeline of drug administration in the GATE-ICH study reinforces the need for future pre-clinical studies to investigate longer time delays between ICH onset and GLC administration than have been assessed thus far. The authors found that GLC administration did not affect incidence of poor outcome at 90 days (assessed via the modified Rankin Scale). However, GLC caused statistically significant reductions in most (growth rate from day 1 to 7, peak PHE, PHE volume, edema extension distance)

but not all (relative PHE) measures of PHE. GLC also significantly lowered blood glucose but did not significantly increase incidence of symptomatic hypoglycemia.

The failure of the GATE-ICH RCT may be partially attributable to an inappropriate patient group. Ischemic [41,47,48] and pre-clinical ICH work have focused on GLC in the context of large strokes with extensive edema, where it is hypothesized to reduce mortality. In contrast, the GATE-ICH study investigated smaller strokes (~9 mL bleed, approximately ~13 μL in rats based on direct brain weight conversion [112]) with a low (2%) mortality rate, which may not have sufficient edema to significantly affect outcomes [41,47,113]. However, this is not concretely known; further work is needed to determine whether bleed size is a factor in GLC efficacy.

Our meta-analysis found that the pre-clinical data provide cautious support for further testing of GLC in ICH. However, due to overall poor study quality, and many STAIR recommendations that remain to be addressed, additional pre-clinical data are needed to provide robust investigation into GLC prior to clinical translation. Future clinical studies should await further pre-clinical work before continuing to investigate GLC in humans.

## Limitations and future directions

This meta-analysis has a number of limitations. First, as considerable data were not accessible, we were forced to extract data from published graphs, and analyze non-parametric data that had been reported using parametric statistics. Some argue that ordinal data can be treated like interval data, if the data are relatively normally distributed [114]. However, due to the lack of available raw data, the normality of our data are not known, and as such, our choices to use parametric summary statistics may have resulted in an overestimation of power [115]. Due to limitations of the data, attempting to generate non-parametric summary statistics was not possible, and likely would have resulted in a larger margin of error.

Second, only six studies from a total of four different labs fit our inclusion criteria and were included in the meta-analysis. Although our analysis did not include pre-ICH dosing regimens, this only excluded one study [96], meaning only seven studies have investigated GLC in experimental ICH. The limited data available on GLC in ICH further limits the power of our study, especially with respect to our Egger regression, and also greatly limited the ability of our meta-regression to detect the influence of other factors on outcome (e.g., stroke severity). Since it is hypothesized that GLC is mostly beneficial in severe strokes [41], future pre-clinical work and reviews should continue to investigate how stroke severity impacts GLC's effect.

Given the effect size established in this meta-analysis, further high quality, large sample size (n > 40) studies are warranted to further characterize the efficacy and mechanisms of GLC after ICH. These studies should include a more diverse demographic by including both female, aged, and co-morbid rodents, and aim to elucidate the therapeutic window of GLC in ICH to inform any future clinical studies.

## Conclusion

In conclusion, we found tentative support for further investigation of GLC in animal models of ICH. Although GLC reduced edema and improved behavioural outcomes, limited dosing data, relatively homogenous animal populations (i.e., younger, healthy male rats only), and concerns with bias and study quality suggest that our data should be interpreted with caution. Additional animal studies should address these concerns before any further clinical work is pursued. Importantly, our summary consists of mostly exploratory work; future confirmatory research is required to more confidently interpret these results [98,99].

## Supporting information

**S1 Appendix. Completed PRISMA checklist.**
(DOCX)

**S2 Appendix. Data files.** All relevant data files and R code are available.
(ZIP)

## Author Contributions

**Conceptualization:** Tiffany F. C. Kung, Cassandra M. Wilkinson, Frederick Colbourne.

**Data curation:** Tiffany F. C. Kung, Cassandra M. Wilkinson, Frederick Colbourne.

**Formal analysis:** Tiffany F. C. Kung, Lane J. Liddle.

**Supervision:** Frederick Colbourne.

**Writing – original draft:** Tiffany F. C. Kung.

**Writing – review & editing:** Cassandra M. Wilkinson, Lane J. Liddle, Frederick Colbourne.

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
