## [Decision Letter · Decision Letter 0]

21 Aug 2023

PONE-D-23-17196A Systematic Review and Meta-Analysis on the Efficacy of Glibenclamide in Animal Models of Intracerebral HemorrhagePLOS ONE

Dear Dr. Kung,

Thank you for submitting your manuscript to PLOS ONE. After careful consideration, we feel that it has merit but does not fully meet PLOS ONE’s publication criteria as it currently stands. Therefore, we invite you to submit a revised version of the manuscript that addresses the points raised during the review process.

1.   Please elaborate more in detail the exclusion criteria of each excluded paper.

2.   On Introduction, page 5, lines 89-94 please elaborate more on the possible mechanisms that may explain the controversial findings in reference to reduction of brain edema and its effects on functional outcome and mortality of ICH. 

3.   Page 11, lines 110-114. Even if many experimental therapies seem to function in animal models of acute stroke, the translational gap should not be forgotten. The therapy options for ischemic stroke in the clinical setting are restricted to thrombolysis and trombectomy in selected patient cohorts.  Please correct and reformulate the text. 

4.   Please also reformulate the ambiguous and misleading sentence on line 113 "Notably, unlike ischemic stroke, serum extrusion is a major cause of edema in ICH, which GLC should not affect". 

We look forward to receiving your revised manuscript.

Kind regards,

Anna-Leena Sirén

Academic Editor

PLOS ONE

Reviewers' comments:

Reviewer's Responses to Questions

**Comments to the Author**

1. Is the manuscript technically sound, and do the data support the conclusions?

Reviewer #1: Yes

2. Has the statistical analysis been performed appropriately and rigorously? 

Reviewer #1: Yes

3. Have the authors made all data underlying the findings in their manuscript fully available?

Reviewer #1: Yes

4. Is the manuscript presented in an intelligible fashion and written in standard English?

Reviewer #1: Yes

5. Review Comments to the Author

Reviewer #1: The author wrote about the effect of Glibenclamide in animal models of intracerebral hemorrhage. The paper exhibits a well-written and comprehensive investigation into the intricate relationship between Glibenclamide and ICH in animal models. The authors have provided an extensive discussion, which significantly contributes to our understanding of this important research area.

Despite the shortage of analyzed studies, the paper demonstrates a commendable level of scientific rigor in its experimental design and data analysis. The methodology employed in this study is robust, and the systematic review procedures are meticulously described, facilitating reproducibility. The authors' discussion showcases a comprehensive understanding of the topic, as they skillfully interpret the experimental results and effectively relate them to existing knowledge in the field.

However, there are several aspects that need to be explained more. We encourage the authors to address the following aspects to strengthen the impact of their study and ensure its relevance to the broader scientific community:

1. In the excluded full papers list, the author mentioned about “four papers were excluded for inappropriate patient populations, and one for using an excluded dosing regimen” Can the author explain more about this reason?

2. The author needs to add an introduction about the importance of defining a new treatment to reduce the edema since the use of diuretics is common in clinical practice. The use of glibenclamide in reducing edema in human patients with ICH and its comparison with common diuretics that are used to treat edema are interesting aspects to be compared.

3. It is mentioned in line 113 "Notably, unlike ischemic stroke, serum extrusion is a major cause of edema in ICH, which GLC should not affect". This line is mismatched with the reason why this systematic review needs to be done. How does the author explain or elaborate on this?

6. PLOS authors have the option to publish the peer review history of their article (what does this mean?). If published, this will include your full peer review and any attached files.

Reviewer #1: No

---

## [Author Response · Author response to Decision Letter 0]

1 Sep 2023

Academic Editor:

1. Please elaborate more in detail the exclusion criteria of each excluded paper.

• We have elaborated and added reasons and citations for each excluded paper in our Search Results section on page 14, lines 324-328. Further, we have added in justification for the exclusion of pre-ICH dosing regimens on page 9, lines 212-215.

2. On Introduction, page 5, lines 89-94 please elaborate more on the possible mechanisms that may explain the controversial findings in reference to reduction of brain edema and its effects on functional outcome and mortality of ICH. 

• We have added this information in on pages 5-6, lines 112-124.

3. Page 11, lines 110-114. Even if many experimental therapies seem to function in animal models of acute stroke, the translational gap should not be forgotten. The therapy options for ischemic stroke in the clinical setting are restricted to thrombolysis and trombectomy in selected patient cohorts. Please correct and reformulate the text. 

• We thank the editor for noting this misleading wording. We have fixed our wording in the original sentence on page 7, lines 146-147, and have added in a separate line clarifying that thrombectomy and thrombolysis remain the only clinically approved therapies for ischemic stroke on page 6, lines 138-140.

4. Please also reformulate the ambiguous and misleading sentence on line 113 "Notably, unlike ischemic stroke, serum extrusion is a major cause of edema in ICH, which GLC should not affect". 

• We have clarified our wording in this sentence on page 7, lines 150-152.

Reviewer #1

The author wrote about the effect of Glibenclamide in animal models of intracerebral hemorrhage. The paper exhibits a well-written and comprehensive investigation into the intricate relationship between Glibenclamide and ICH in animal models. The authors have provided an extensive discussion, which significantly contributes to our understanding of this important research area.

Despite the shortage of analyzed studies, the paper demonstrates a commendable level of scientific rigor in its experimental design and data analysis. The methodology employed in this study is robust, and the systematic review procedures are meticulously described, facilitating reproducibility. The authors' discussion showcases a comprehensive understanding of the topic, as they skillfully interpret the experimental results and effectively relate them to existing knowledge in the field.

However, there are several aspects that need to be explained more. We encourage the authors to address the following aspects to strengthen the impact of their study and ensure its relevance to the broader scientific community:

1. In the excluded full papers list, the author mentioned about “four papers were excluded for inappropriate patient populations, and one for using an excluded dosing regimen” Can the author explain more about this reason?

• We have elaborated and added reasons and citations for each excluded paper in our Search Results section on page 14, lines 324-328. Further, we have added in justification for the exclusion of pre-ICH dosing regimens on page 9, lines 212-215.

2. The author needs to add an introduction about the importance of defining a new treatment to reduce the edema since the use of diuretics is common in clinical practice. The use of glibenclamide in reducing edema in human patients with ICH and its comparison with common diuretics that are used to treat edema are interesting aspects to be compared.

• We have added in a paragraph elaborating on the treatment of edema in ICH and included the most common clinical management methods on page 7, from lines 153-169. As HTS and mannitol were not investigated in our systematic review, the section was kept brief, with references cited to guide readers who may be interested in reading about these therapies in more detail.

3. It is mentioned in line 113 "Notably, unlike ischemic stroke, serum extrusion is a major cause of edema in ICH, which GLC should not affect". This line is mismatched with the reason why this systematic review needs to be done. How does the author explain or elaborate on this?

• We thank the reviewer for noting our confusing wording. Highlighting serum extrusion as something that occurs in ICH but not in ischemic stroke was meant to reinforce the importance of original research on GLC specifically in the context of ICH, rather than the meta-analysis itself. Due to key differences between stroke subtypes, it remains important that ischemic stroke results not be used to justify the use of GLC in ICH patients. We have elaborated on this in the manuscript on page 7, lines 147-152.

---

## [Editor Report · Decision Letter 1]

11 Sep 2023

A systematic review and meta-analysis on the efficacy of glibenclamide in animal models of intracerebral hemorrhage

PONE-D-23-17196R1

Dear Dr. Kung,

We’re pleased to inform you that your manuscript has been judged scientifically suitable for publication and will be formally accepted for publication once it meets all outstanding technical requirements.

Kind regards,

Anna-Leena Sirén

Academic Editor

PLOS ONE
---

## [Editor Report · Acceptance letter]

18 Sep 2023

PONE-D-23-17196R1 

A systematic review and meta-analysis on the efficacy of glibenclamide in animal models of intracerebral hemorrhage 

Dear Dr. Kung:

I'm pleased to inform you that your manuscript has been deemed suitable for publication in PLOS ONE. Congratulations! Your manuscript is now with our production department. 

Kind regards, 

on behalf of

Dr. Anna-Leena Sirén 

Academic Editor

PLOS ONE